# Development of a Fluorine-Free Polymer-Assisted-Deposition Route for YBa$_2$Cu$_3$O$_{7-x}$ Superconducting Films

**Mircea Nasui [1]** , **Ramona Bianca Sonher [1], Traian Petrisor Jr. [1],*, Sorin Varodi [1], Cornelia Pop [2],
Lelia Ciontea [1] and Traian Petrisor [1]**

[1]   Centre for Superconductivity, Spintronics and Surface Science, Physics and Chemistry Department,
     Technical University of Cluj-Napoca, Str. Memorandumului, Nr. 28, 400114 Cluj-Napoca, Romania;
     mircea.nasui@chem.utcluj.ro (M.N.); ramona.mos@chem.utcluj.ro (R.B.S.); Sorin.Varodi@phys.utcluj.ro (S.V.);
     lelia.ciontea@chem.utcluj.ro (L.C.); traian.petrisor@phys.utcluj.ro (T.P.)
[2]   Institut de Ciència de Materials de Barcelona, ICMAB—CSIC, Campus UA Barcelona,
     E-08193 Bellaterra, Catalonia, Spain; cpop@icmab.es
*    Correspondence: traian.petrisorjr@phys.utcluj.ro

**Abstract:** Polymer assisted deposition (PAD) was used as an environmentally friendly, non-fluorine, growth method for superconducting YBa$_2$Cu$_3$O$_{7-x}$ (YBCO) films. The kinetics of the thermal decomposition of the precursor powder was studied by thermogravimetry coupled with mass spectrometry (TG-QMS). YBCO films were spin coated on (100) SrTiO$_3$ (STO) single crystalline substrates, followed by a single step thermal treatment under wet and dry O$_2$ and O$_2$/N$_2$ mixture. The as-obtained films were epitaxially grown having a [001]YBCO‖[001]STO out-of-plane epitaxial relationship and exhibited good superconducting properties with $T_c$ ($R = 0$) > 88 K, transition widths, $\Delta T \approx 2$ K and critical current densities as high as 2.3 MA/cm$^2$ at 77 K and self magnetic field.

**Keywords:** fluorine-free coating solution; polymer assisted deposition; YBCO superconducting films

## 1. Introduction

High-temperature superconducting (HTS) YBa$_2$Cu$_3$O$_{7-x}$ (YBCO) films have been extensively studied mainly due to their potential applications in superconducting electric power applications such as electrical energy transportation, motors, generators, magnets for nuclear fusion, magnetic energy storage, etc. [1]. Chemical solution deposition (CSD) route has proven to be a valid alternative in the formation of complex oxide thin films, that can also be scaled-up for long length superconducting tape fabrication [2,3]. The CSD of YBCO films involves four steps: precursor synthesis, deposition, decomposition of organic precursors (pyrolysis) and film crystallization. Understanding and controlling the thermal decomposition of the precursor solution and its conversion to epitaxial YBCO thin films are critical for reproducible manufacturing of uniform, high-performance HTS tapes.

Up to date the trifloroacetate-based precursor solution for YBCO thin film growth has been extensively investigated because it can produce superconducting films with excellent current carrying capabilities [2,4]. However, one of the major drawbacks of this method is the liberation of toxic HF during the YBCO pyrolysis/crystallization steps [5]. Continuous efforts are being carried out in the development of low-fluorine (see for example [6,7]) and fluorine-free chemical solution deposition processes [8–12]. These approaches consist of dissolving carboxylates, such as acetates, propionates, etc., or β-diketonates, as metal precursors in a common solvent, usually organic acids, and combining the solutions to yield the desired stoichiometry. The combination of short chain carboxylates, such as acetates, and strongly chelating β-diketonates, or propionates of all the metals dissolved in propionic

acid ($CH_3CH_2COOH$) has been successfully used [13]. The use of chelating agents leads to the formation of monomeric compounds and the lower organic content of the precursors reduces excessive weight loss and shrinkage during the film decomposition, suppressing problems with film cracking. The most demanding requirements for YBCO thin film growth are on one hand, the need to stabilize highly concentrated solutions, which enable the growth of thick, single coated layers. On the other hand, residual carbon needs to be eliminated from the final films, to preserve good superconducting transport properties of the as-obtained films. For example, triethanolamine is used to increase the solubility of acetates in methanol/water, since the –OH groups provided by TEA molecules can be deprotonated and formation of mixed carboxylate-TEA complexes [14]. The formed complex takes place ensures the stability and homogenous distribution of the metal ions. Besides, the propionic acid is also a promising solvent for dissolving acetates because it can be obtained high concentrations and has excellent wetting behavior at the substrate surfaces.

However, fluorine-free precursor solutions method has a major drawback. The formation of $BaCO_3$ phases during the heat treatment can degrade the superconductivity of the YBCO films [15]. Several authors [16,17] reported on the successful decomposition during the heat treatment and obtained YBCO films with high critical current. These studies related that $BaCO_3$ can react with CuO to generate the $BaCuO_2$ phase which then reacts with $Y_2O_3$ to form YBCO, thus achieving the elimination of the $BaCO_3$ in the final films. This process takes place at high temperatures in low oxygen partial pressure environment.

Recently, a polymer-based chemical deposition method has been developed, which is becoming a promising alternative for the synthesis of oxide thin films [18], including YBCO layers [19–21]. Dense, epitaxial oxide thin films were initially obtained by Jia et al. [22] using the polymer-assisted deposition (PAD) method showing a large improvement in the crystallinity and structural quality with respect to most of the previous CSD approaches [23].

In the PAD thin film synthesis, as opposed to other CSD methods, a polymer able to bind and stabilize the metal precursors is used in the preparation of the precursor coating solution. The polymer binded metals ion are homogenously distributed within the coating solution, ensuring the growth of highly uniform films. After preparation, the metal-polymer solution is filtered and the metal ion concentration is determined by inductively coupled plasma atomic emission spectroscopy (ICP-AES). This last step is also a characteristic of the PAD thin film growth method. Metal nitrates, chlorides or acetates and ethylenediaminetetraacetic acid (EDTA), polyethyleneimine (PEI) etc. [24] as chelating agents were used for the preparation of the coating solutions. EDTA is known to form very stable complexes with a wide range of different cations, Cu(II), Ni(II), Fe(III), Cr(III), Co(II), Mn(II), Ba(II), and Ir(IV) [25,26]. Alternatively, PEI is also a promising polymer for its excellent wetting behavior of substrate surfaces. Moreover, PEI can establish hydrogen bonds and electrostatic interactions with an anionic metal complex $[EDTA-Metal]^{n-}$ [27].

In this paper, an environmentally friendly polymer based method was used to synthesize YBCO thin films. The coating solution was prepared by dissolving Y-, Ba-, Cu- nitrates in EDTA using water as a solvent. The pH was adjusted by adding PEI. This coating solution can be prepared at room temperature, by simple mixing of the corresponding (Y-, Ba-, Cu-) metal ions with suitable chelating agent and polymer. The final coating solution was stable at room temperature for 2 months without any visible changes. The final films have been characterized in terms of structure, morphology, and superconducting transport properties.

## 2. Materials and Methods

The YBCO precursor solution was prepared using Y-, Ba- and Cu- nitrates. The nitrates were separately dissolved in water, EDTA, and PEI with $M_w = 70,000$. The electrostatic interactions between the protonated amino groups of PEI and the $[EDTA-Metal]^{n-}$ complex are crucial for the further successful deposition of a homogeneous film [18]. The solution pH was adjusted to a value of five with PEI. PEI addition also improves the precursor solution viscosity, wettability and stability [19,21].

In order to remove uncoordinated species (cations/anions), the PAD precursor solution is filtered using Amicon ultrafiltration unit containing a PM10 ultrafiltration membrane. The final metal concentrations of the individual nitrate solutions were determined by inductively coupled plasmaatomic emission spectroscopy (ICP-AES), resulting in the following concertation values: 81 mg/L for Cu, 88 mg/L for Ba and 77.5 mg/L for Y, respectively. Taking into account the above ICP-AES data, YBCO coating solution was obtained by mixing the nitrate solutions in the appropriate amount to obtain the desired 1:2:3 stoichiometry for Y:Ba:Cu. The dynamic viscosity of the solutions was measured at 20 °C with Brookfield DV1 Cone/Plate Rheometer and was adjusted in the range of 4–5.5 mPa·s by adding PEI, in order to obtain optimal deposition conditions for spin coating.

The thermal decomposition mechanism of the precursor powder was investigated by thermogravimetric analysis (TG) coupled with mass spectrometry (MS). The YBCO precursor powder was prepared by drying the coating solution in vacuum and subsequent heating at a temperature of 700 °C with a heating rate of 5 °C/min. The total metal ion concentration in the final precursor solution is 0.4 M. The solutions were then coated on the $10 \times 10$ mm$^2$ SrTiO$_3$ (001) (STO) substrates by spin coating.

The coated films were subjected to a single step thermal treatment as presented in Figure 1. The pyrolysis step between 550–650 °C for 60 min. has been performed to ensure the complete decomposition of the organic matter. Film crystallization has been performed at 835 °C for two hours both in humid oxygen, and nitrogen environment, at a rate of 10 °C/min. The film was cooled down to 450 °C in the same atmosphere at a rate of 10 °C/min, held at this temperature for one hour in oxygen atmosphere for oxygenation and orthorhombic phase formation. Finally, the film was cooled at room temperature.

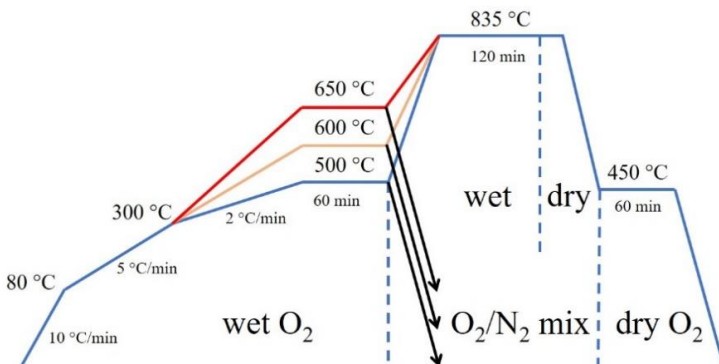

**Figure 1.** The heat treatment profiles of the YBCO films with different holding temperatures from 500 to 650 °C at the pyrolysis step.

The structural properties of the as-obtained YBCO films were investigated by X-ray diffraction. The X-ray $2\theta/\omega$ scan was performed with a Bruker D8 Discover X-ray diffractometer (XRD, Billerica, MA, USA) using Cu K$\alpha$ radiation. The thickness of the YBCO film was evaluated via transmission electron microscopy (TEM) using a JEOL JEM-2100F Field Emission Microscope (Tokyo, Japan), HELIOS NanoLab600 Focused Ion Beam (FIB) was used to cut a 60–100 nm thick lamella from the prepared sample (other dimensions of the lamella are 5–7 μm in length and 3–4 μm in height) for the TEM investigation. The electrical characterization ($R(T)$ and $J_c(T)$), were performed using the four-point technique and SQUID magnetometery. The critical current density, $J_c$, was derived from $M$(H) curves using the Bean critical state model.

## 3. Results

The most important aspect of the PAD process is the thermal decomposition of the polymer. The fact that the metals remain homogenously mixed until the polymer is removed allows the formation of high-quality complex metal-oxide films.

The TG analysis of the PAD-precursor powder was performed in inert and oxidative atmospheres in order to define the decomposition steps. The TG curves of the precursor powder annealed in nitrogen, air and dry/humid oxygen are shown in Figure 2a. Three steps are observed in the following temperature ranges: 23–180 °C (1st stage), 180–430 °C (2nd stage) and 430–600 °C (3rd stage). The three broad steps are observed in each case, but much more clearly defined for wet $O_2$ atmosphere. This suggests the formation of intermediate hydrates with masses and stability ranges dependent on the atmosphere. Humid oxygen atmosphere was chosen for MS analysis.

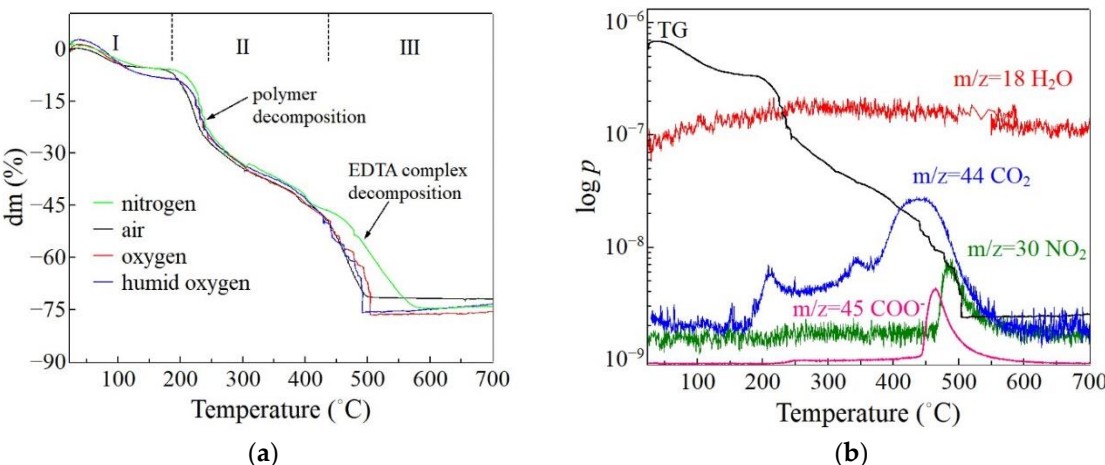

**Figure 2.** (**a**) TG curves in oxidative and inert atmosphere; (**b**) MS curves of the precursor powder heated at a rate of 5 °C/min (TG curve was added for reference).

The first weight loss corresponds to water and coordinated solvent molecules evaporation. The mass spectrum, Figure 2b, of the gas produced in the first stage decomposition contained the $m/z = 18$ signal, which corresponds to water. The second step (in the temperature range of 180–430 °C) has a mass loss of 45% due to the decomposition of the polymer. Water and carbon dioxide ($m/z = 44$) is detected by mass spectrometry at the same time. This result is in good agreement with the thermal decomposition of the PEI-polymer [14]. The third step (430–600 °C) has a mass loss of 30% due to the decomposition of the metal EDTA complex. The MS spectra have revealed that the evolved gases correspond to fragments with $m/z = 45$ characteristic for $COO^-$, $m/z = 30$ for $NO_2$, and $m/z = 44$ for $CO_2$. From the above MS analysis, we can propose that during the first stage only the solvent decomposes while during the second and third stages, both polymer and EDTA complex decompose.

Several heat treatment parameters make an impact on the final properties of CSD-YBCO film. To establish the most important factors influencing the removal of organic species and the subsequent formation of oriented superconducting YBCO layer, the pyrolysis step was investigated separately from the crystallization and oxygenation steps in the YBCO layer growth. Accordingly, three samples were prepared by quenching at room temperature after a 60 min. pyrolysis at 500, 600 and 650 °C. The quenching profile used is show in Figure 1.

The XRD pattern of the film quenched at 500 °C, Figure 3a, exhibits no film peaks confirming that the film is in an amorphous state. Many studies show that during the thermal decomposition of the polymer (PEI) between 23–550 °C, the polymer is effectively molten providing an amorphous state [28].

The thermal stability of PEI provides a great advantage, as it promotes a controllable decomposition until the YBCO film formation. It prevents the formation of individual oxides when a multi-cationic compound is required. When the film is heated up to 600–650 °C, (00*l*) YBCO diffraction peaks may be observed and with temperature increase, the intensity of YBCO phase corresponding peaks increases. Annealing above 600 °C a complete decomposition of the polymer-organic complex in volatile species, without any remains of organic material in the sample, as was demonstrated by TG analysis performed into oxygen. Furthermore, the slow decomposition of the polymer, at a final temperature close to the

crystallization temperature of the YBCO promotes the formation of the YBCO film at a very slow rate, close to thermodynamic equilibrium conditions, leading to high crystallinity films.

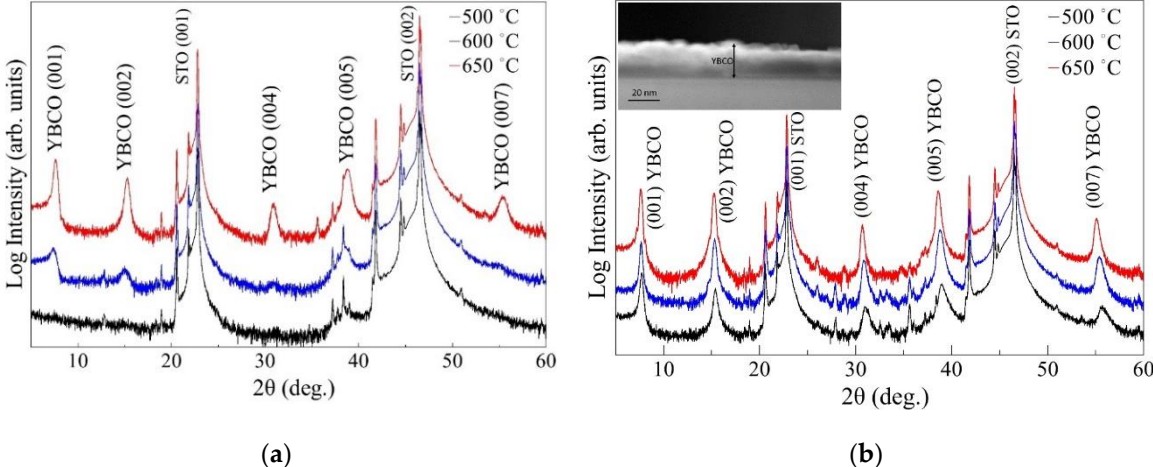

**Figure 3.** XRD 2θ/ω patterns of the (**a**) films quenched with different holding temperatures at the pyrolysis step and (**b**) final YBCO films (after crystallization at 835 °C) with different pyrolysis temperatures.

The 2θ/ω XRD patterns of the YBCO films deposited on (001) STO substrate is presented in Figure 3b. As it can be seen, the XRD pattern exhibits YBCO reflections pertaining only to the (00l) family, indicating that the YBCO film is epitaxially grown on (001)STO with a [00l]YBCO//[00l]STO epitaxial relationship between the substrate and the film. No grains with the a/b-axis oriented perpendicular to the substrate are observed. In order to quantify the effects of the pyrolysis temperature on the structural properties of the final YBCO thin films, several quantities were determined and summarized in Table 1.

**Table 1.** Dependence of structural parameters of YBCO films on the pyrolysis temperature.

| Pyrolysis Temperature (°C) | 550 | 600 | 650 |
|---|---|---|---|
| $c$ lattice parameter (Å) | 11.553 | 11.604 | 11.665 |
| $I_{(007)\ \text{YBCO}}/I_{(002)\ \text{STO}}$ (%) | $2.41 \times 10^{-5}$ | $7.96 \times 10^{-5}$ | $11.1 \times 10^{-5}$ |
| FWHM (°) | 0.836 | 0.659 | 0.467 |

The $c$ lattice parameter, determined from the position of the YBCO (005) peak position and using Bragg's relation, of the films is found to increase with the increase of the pyrolysis temperature, reaching a maximum value of 11.665 Å in the case of the 650 °C pyrolyzed film, close to the bulk value of 11.6802 Å of the orthorhombic YBCO lattice. Additionally, a basic peak profile analysis in terms of relative peak intensity ($I_{(007)\ \text{YBCO}}/I_{(002)\ \text{STO}} \cdot 100(\%)$) and full-width at half-maximum (FWHM) was performed on the (007) YBCO peak. The FWHM was found by performing a gaussian fit of the respective peaks. The increase of the YBCO peak intensity, relative to the substrate STO (002) peak, accompanied by a decrease of the FWHM was recorded as a function of pyrolysis temperature increase. In the inset of Figure 3b we present the TEM image of a section through a STO substrates with YBCO layer. The presented TEM image indicates that the film is dense without any pores or cracks, while the film thickness was evaluated around 20 nm. The increase of the c-axis lattice parameter with increasing the pyrolysis temperature may be associated with a relaxation of the crystal structure which, for the reported thickness, is in a state of tensile strain due to the higher lattice parameter of STO, 3.905 Å, than the in-plane YBCO parameters ($a$ = 3.823 Å, $b$ = 3.887 Å). The presence of strain for lower pyrolysis temperature may lead to a cation disorder [22] which accounts for the reduced YBCO peak intensity

and higher FWHM values. On the other hand, at low temperatures the observed structural features can also be ascribed to an incomplete precursor decomposition, which leads to carbon poisoning of the lattice of the crystalized film.

In Figure 4 the temperature variation of the normalized electrical resistances of the three samples are presented. The normalization was performed with respect to the resistance at around 296 K. Recently, similar metallic temperature dependence of the electrical resistance has been reported in an intermediate temperature range, 50–200 K, for other oxide heterostructures [29].The variation of the critical transition temperature, $T_c$, as a function of the pyrolysis temperature is shown in the inset of Figure 4.

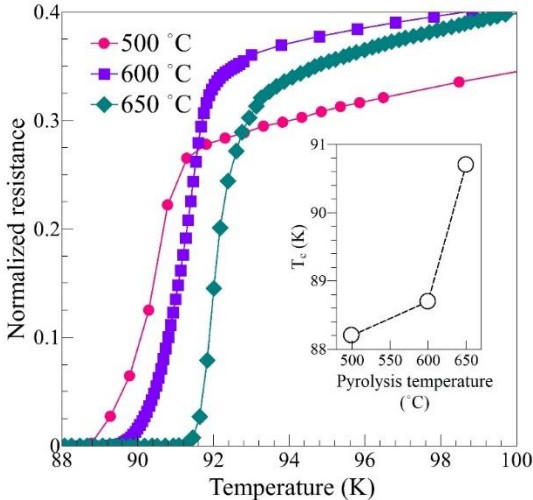

**Figure 4.** Normalized $R(T)$ measurements for the crystallized films, treated at different pyrolysis temperatures; (inset) Variation of the critical superconducting transition temperature $T_c$ as a function of pyrolysis temperature.

The $T_c$ of the superconducting films increases with the increase of pyrolysis temperature, from 88.2 K for the sample pyrolyzed at 500 °C to 90.7 K in the case of the 650 °C treated film. The width of the transition was determined from the data as $\Delta T = T_{90} - T_{10}$, where $T_{90}$, $T_{10}$ represent the temperatures corresponding to the 90% and 10% of values of the normalized resistance at the onset of the superconducting transition. For all three samples $\Delta T$ was situated in a narrow, 2–2.1 K, interval. The determined $T_c$ values and transition widths are in good agreement with those reported for other PAD/CSD YBCO films [12,20,21]. Cation disorder present in the strained films pyrolyzed at lower temperatures determine a reduction of $T_c$ [30]. However, as strain is relaxed cation disorder is reduced and hence an increase of the critical temperature is recorded as a function of pyrolysis temperature.

The temperature dependence of the critical current density of the three samples is presented in Figure 5. It may be observed that in a wide temperature range the highest measured $J_c$ is that of the sample pyrolyzed at 650 °C. This is true at liquid nitrogen temperature of 77 K as well, where the $J_c$ for the afore mentioned sample is 2.3 MA/cm$^2$. However, in the 82–84 K interval there is a crossover. In the high temperature regime, the sample heat treated at 550 °C has the highest $J_c$ value. Also, the as reported values of $J_c$, are comparable with the values reported in other PAD/CSD grown layers [12,20,21]. To further investigate this behavior a fit of the $J_c(T)$ dependencies was performed. In the intermediate temperature range 10–50 K, the critical current density is known to decay exponentially associated with the weak collective pinning model [31], following the relationship:

$$J_c(T) = J_{c0} \exp(T/T_0) \tag{1}$$

where $J_{c0}$ is the critical current density extrapolated at 0 K, while $T_0$ is related to the vortex pinning potential present within the film. For the present samples, good agreement was found between the measurement and the above relationship, in the 20–50 K range, in which fit was performed. The dependencies of the two fit parameters, $J_{c0}$, $T_0$, on the pyrolysis temperature is illustrated in the inset of Figure 5.

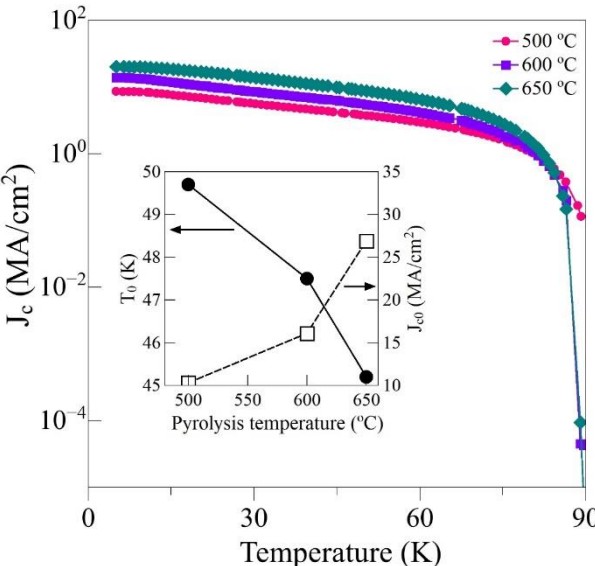

**Figure 5.** Temperature dependence of the critical current density $J_c(T)$ of the YBCO thin films treated at different pyrolysis temperatures; (inset) Variation of the fitting parameters $J_{c0}$ and $T_0$ as a function of pyrolysis temperature (see text).

It may be observed that, while the $J_{c0}$ increases as a function of the annealing temperature, $T_0$ decreases as the temperature is increased. This variation may be understood from the structural and micro-structural evolution of the films with increasing pyrolysis temperature. As synthesized in Table 1, as the pyrolysis temperature increases the structural quality of the resulting films increases. Good structural properties determine a higher $J_{c0}$. Cation disorder present in the low temperature treated samples determine regions in which the superconducting order parameter is suppressed leading to point defect flux pinning sites [32]. Pinning reduces vortex thermal activation, which in turn determines a weaker temperature dependence of $J_c$, hence higher $T_0$ values. Higher $T_0$ values for both the 500 °C and the 600 °C films, may thus be associated with the presence of a higher number of lattice defects within these samples. The high temperature cross-over observed for the $J_c(T)$ dependencies is therefore due to the effective pinning sites present in the 500 °C pyrolyzed film, related to cation disorder.

## 4. Conclusions

Our investigations have demonstrated that the proposed fluorine-free PAD process is very promising to obtain YBCO films with good superconducting properties in a single step thermal treatment. Good superconducting properties were recorded for the elaborated films, with critical temperature reaching 90.7 K and critical current density of 2.3 MA/cm$^2$ at 77 K in self field, being reported. Pyrolysis temperature was demonstrated to play a crucial role in determining the physical properties of the final layers. The main issue affecting the proposed approach is the relatively low final film thickness. This issue may be overcome in the future by multiple subsequent deposition and/or by fine tuning the balance between precursor solution viscosity, polymer quantity and metal ion concentration. Alternatively, the proposed process may be used such as seed layer for further

YBCO growth, or as a versatile, low-cost, environmentally friendly solution for high-temperature superconducting electronics applications.

**Author Contributions:** Conceptualization: M.N. and L.C.; formal analysis: T.P.J., M.N., and C.P.; investigation: M.N., R.B.S., C.P., S.V., and T.P.J.; supervision: M.N., L.C., and T.P.; funding acquisition: T.P.J., T.P., and M.N.; visualization: M.N. and T.P.J., writing—original draft: M.N., T.P.J., and C.P.; writing—review and editing: M.N., T.P.J., and T.P. All authors have read and agreed to the published version of the manuscript.

**Funding:** This work was supported by MRI-UEFISCDI through PN-III-P1-1.1-TE-2016-2465-SuperMagSense contract No. 80/02.05.2018, project 21 PFE-2018 and contract No. 1991 Internal Competition CICDI-2017.

**Acknowledgments:** Special thanks are given to ICMAB (Institut de Ciència de Materials de Barcelona) for their scientific services and also to E. Ware from the Imperial College London, United Kingdom, for the TEM investigation.

**Conflicts of Interest:** The authors declare no conflict of interest. The funders had no role in the design of the study; in the collection, analyses, or interpretation of data; in the writing of the manuscript, or in the decision to publish the results.

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
