# Peer review of "Development of a Fluorine-Free Polymer-Assisted-Deposition Route for YBa2Cu3O7−x Superconducting Films"

_coatings, doi:10.3390/coatings10100966_

Round 1
Reviewer 1 Report
Reviewer report on Manuscript Draft entitled ‘Development of a fluorine-free polymer-assisted-deposition route for YBa2Cu3O7-x superconducting 3 films'.
In this research polymer assisted deposition (PAD) was used as an environmentally friendly, non fluorine, growth method for superconducting YBa2Cu3O7-x (YBCO) films. The kinetics of the thermal decomposition of the precursor powder was studied by thermogravimetry coupled with mass spectrometry (TG-QMS). These investigations have demonstrated that the proposed fluorine-free PAD process is very promising to obtain YBCO films with good superconducting properties in a single step thermal treatment. The proposed process may be used such as seed layer for further YBCO growth, or as a versatile, low-cost, environmentally friendly solution for high-temperature superconducting electronics applications.
Research presented in this manuscript is interesting and well-illustrated by results. Research fits with the scope of journal. Therefore, the manuscript eventually can be published after some corrections and improvements:
Results presented in Result and discussion part are not well discussed and compared with that registered by other researchers, therefore this part of manuscript should be improved by comparison here reported with that observed by other researchers.
Rather good resistance dependence on temperature was determined (figure 4), therefore, resistance dependencies on temperature (figure 4) should be compared with that of some other conducting materials (TiO2-x/TiO2-structure based ‘self-heated’ sensor for the determination of some reducing gases. Sensors 2020, 20, 74.) and discussed
Author Response
- The properties of the as-obtained YBCO thin films are comparable with those of films reported in the articles cited in the bibliography. However, explicit references have been made to articles which obtained comparable values of the two most relevant superconducting electric transport properties, critical transition temperature (rows 222,223) and critical current density (rows 231, 232, 233).
- Indeed the suggested article shows a similar temperature dependence of the resistance of the TiO2-x/TiO2 structure, in the 50 - 200 K interval. This observation has been included in the text (rows 209, 210) together with the suggested reference.
Reviewer 2 Report
In line 44 the authors briefly hints to the previous CSD approaches. This section should be expanded to include the most significant works in the field and to highlight the main differences between CSD and PAD.
Quality of Figure 1 should be enhanced. All figures should be expressed with the same symbols in caption and in text (i.e. Figure 1 and not Fig. 1).
In line 91, the authors refer using a FIB to cut a thick lamella. Please add how thick was the lamella and add images of this. There is a hint of an image in Figure 3 b, however this is too small and of a very poor resolution. Please also add an explanation of why FIB was necessary and how did you process the results of FIB.
In line 152, the authors stated that ‘several quantities were determined and summarised’. Could the author refer back to the methodology to explain how the quantities summarised in Table 1 were determined?
In line 163 the authors mention about TEM results. Why HRTEM has not been included in the methodology also, why there is not images referring to the TEM results?
Author Response
- The subject of fluorine-free CSD of YBCO thin films has been extensively enhanced (rows 40-63), including the appropriate relevant references. In rows 69, 71-74 the main difference between PAD and CSD methods is elaborated. However, PAD growth of YBCO thin films is not a very well documented research topic. The available articles have been cited Ref. 19-21, together with other articles reporting on the growth of oxide films by PAD, Ref. 22, 24, 26.
- Figure 1 has been updated and all figures are referred to as "Figure X".
- The geometrical dimensions of the FIB lamella has been include in the text, together with an indication that the lamella is required for the TEM investigation (rows 123-125). Unfortunately, an image of the lamella is not available.
- The detailed methods for obtaining the parameters in Table 1 have been described in the text (rows 188,189 for the c lattice parameter, row 192 for the film/substrate relative diffraction intensities, rows 193, 194 for the FWHM of the (007) YBCO peaks).
- There was an error when referring to HRTEM. The TEM investigation consisted of the image presented in the inset of Figure 3b, from which the thickness of the YBCO film was evaluated, together with an overview of the microstructure of the film, "dense, without any pores or cracks".
Reviewer 3 Report
In this paper Authors used a polymer assisted deposition (PAD) as an environmentally friendly, non-fluorine, growth method for superconducting YBa2Cu3O7-x (YBCO) films. The kinetics of the thermal decomposition of the precursor powder was studied by thermogravimetry coupled with mass spectrometry (TG-QMS). YBCO films were spin coated on (100) SrTiO3 (STO) single crystalline substrates, followed by a single step thermal treatment under wet and dry O2 and O2/N2 mixture.
The originality the concepts, the significance and the methods are good. In my opinion the technical treatment is plausible and free of technical errors. Below I presented some remarks that came to my mind during reading:
- I think that in the introduction, the authors should present in more depth the research of other researchers dealing with this subject. I propose to describe in more detail the focus of the works related to the development of fluorine-free chemical solution deposition processes [6-10]. This will give a more complete picture to people not fully related to the topic of the paper, who can also view it.
- Figure 1. The figure should be placed in the chapter in which it was mentioned for the first time.
- In Conclusions Authors should also present the findings also highlighting current limitations of their study, and briefly mention some precise directions that they intend to follow in their future research work.
- “Author Contributions” and “Conflicts of Interest” needs to be improved - now includes extracts from the authors' instructions.
Author Response
- The fluorine-free CSD of YBCO thin films part has been extensively enhanced (rows 39-63), as suggested, including additional relevant references.
- Figure 1 has been moved to Section 2.
- The main drawback of the proposed approach is the low thickness of the film, i.e. 20 nm. Two suggestions for future work are given, rows 265-267, consisting of multiple depositions and tuning the balance between precursor solution viscosity, polymer quantity and metal ion concentration. These two approaches are intended to increase the final thickness of the YBCO thin films.
- The "Author Contribution" and "Conflicts of Interest" have been updated.